# Synthesis of NiAl Intermetallic Compound under Shock-Wave Extrusion

**DOI:** 10.3390/ma15176062

**Published:** 2022-09-01

**Authors:** Andrey Malakhov, Denis Shakhray, Igor Denisov, Fanis Galiev, Stepan Seropyan

**Affiliations:** 1Merzhanov Institute of Structural Macrokinetics and Materials Science of Russian Academy of Sciences, 142432 Chernogolovka, Russia; 2Institute of Problems of Chemical Physics of Russian Academy of Sciences, 142432 Chernogolovka, Russia

**Keywords:** exothermic reaction, NiAl, shock-wave extrusion, microstructure, cumulative flow

## Abstract

This paper presents the implementation of the first stage of a study on the synthesis of the intermetallic compound in the Ni-Al system under shock-wave extrusion (SWE). A method was developed and experiments involving SWE of the reactive Ni–Al powder mixture were carried out. As a result, it was possible to obtain up to 56 vol.% of the final product and achieve 100% synthesis of NiAl. The results of metallographic analysis indicate that the process of high-velocity collapse of the tube created conditions for the formation of a cumulative flow, which directly affects the phase formation in NiAl. It was shown that the presence of the central hole in the powder sample reduced the effect of the Mach stem on the homogeneity of the NiAl structure. It was also determined that with a central hole with a 5 mm diameter, the effect of the Mach stem could not be observed at all. The goals of further studies are achieving 90–100 vol.% of the final product and reducing the porosity in the final product. Preliminary experimental studies have shown great potential for SWE to produce composite metal–intermetallic materials.

## 1. Introduction

Currently, there is a need for materials with high wear and corrosion resistance under high service temperatures [1]. Among these materials are intermetallic compounds (ICs). They are widely used in civil engineering and in military industry due to their high hardness, chemical resistance and melting point [2]. One of the most widely used ICs is NiAl due to its high melting point (1638 °C) [3,4,5,6]. However, NiAl has low ductility at room temperature. This limits its wider application as a structural material [7]. The problem can be partially solved by adding various additives to the initial mixture [7,8,9,10].

There are three main methods for producing ICs: liquid-phase sintering, solid-phase synthesis and gas-phase synthesis. Liquid-phase sintering includes casting and imbibition [3,11]. Gas-phase synthesis includes physical/chemical deposition [12] and magnetron sputtering technique [13]. Solid-phase synthesis includes self-propagating high-temperature synthesis (SHS) [14], thermal explosion [15], hot gas extrusion [16], spark plasma sintering [17], rolling [18,19], hot isostatic pressing [20], mechanical alloying [21] and shock-wave loading (SWL) [22,23,24,25,26,27,28,29].

All these technologies are used for the synthesis of NiAl, and each of them has its own advantages and disadvantages. For example, despite high productivity, the liquid-phase methods do not allow for control of the process of NiAl synthesis, which leads to deviations in the chemical composition and properties of the NiAl. On the other hand, gas-phase synthesis allows for control of the process of production but has low productivity. Solid-phase synthesis is a highly efficient method for producing NiAl.

The solid-phase methods such as SWL, hot gas extrusion and rolling cause substantial deformation and distortion of the crystal structure, which improves the mechanical properties of NiAl [22].

Among all of the solid-phase methods, SWL causes the highest level of deformation of the crystal structure due to the high pressure (up to 170 GPa) produced by the shock waves [26,27,28,29]. SWL involves high local stress and temperature gradients, forced mass transfer, mechanical activation, etc. Due to these factors, the method makes it possible to produce a nonporous material [22,23,24]. The SWL of powder mixtures is performed by using either a flat or a cylindrical recovery ampoule [24].

It is well-known that it is difficult to create the same loading conditions in the center and the rim of the powder sample in cylindrical recovery ampoules. They are caused by the Mach stem (the region of high pressure and temperature), which is formed along the axis of the powder sample [25]. On the other hand, using flat ampoules creates the same loading conditions in the center and the rim of the powder sample; however, it is still not possible to completely exclude heterogeneity of the structure, cracks and pores in the synthesis products. These disadvantages of recovery ampoules are linked to the way the shock waves propagate in the ampoule and in the powder sample. 

The synthesis products formed after the SWL of Ni–Al powder mixture have been studied by many authors [22,23,24,25,26,27,28,29,30,31]. They have found that the SWL of Ni–Al powder mixture both in cylindrical and in flat ampoules leads to the formation of NiAl intermetallic compound [30,31,32,33].

It is understood that the development of mechanical engineering requires producing not just ICs but the composite material that would combine a metallic and intermetallic layer. However, due to the significant difference in the properties of ICs and metals, producing such a composite material is associated with significant difficulties. Therefore, a combination of several technologies, such as SHS, SWL, and extrusion, is promising. With this combination of processes, the extrusion of ICs should take place directly at the SWL and SHS stages. In this case, homogeneous material of the desired size with minimal porosity can be made. The extrusion of reactive material through a forming hole at a detonation velocity of an explosive is proposed to be called shock-wave extrusion (SWE).

Thus, the aim of this paper is to conduct studies of the synthesis behavior of Ni–Al powder mixture during SWE and to establish the influence of SWE parameters on the formation of NiAl.

## 2. Materials and Methods

### 2.1. Powder Sample Preparation

Experiments were conducted with an Ni–Al powder mixture. This powder mixture consisted of commercially available nickel powder PNK-UT3 (d~18 µm) and aluminum powder ASD-1 (d~20 µm) at the equimolar ratio. The initial powders were mixed in a tumbling drum mixer for 2 h at 30 rpm. The mass ratio between the balls and powder was 5:1. The milled powders were pressed with uniaxial pressure in a steel die into cylindrical samples (Figure 1) with relative densities in the range of 65–70% of the maximum theoretical value. In [34], experiments with the SWL of Ni–Al powder samples (10 mm in diameter and height) were conducted. The results of the experiments showed that it is possible to initiate an exothermic reaction in the powder sample. Given this, the size, composition, and properties of the cylindrical samples were taken from [34]. In addition, two through holes were drilled (2.5 mm and 5 mm in diameter) in the center of both Ni-Al powder cylindrical samples.

### 2.2. The Cylindrical Recovery Ampoules for SWE

Figure 2a shows a photograph of the cylindrical recovery ampoules before SWE. The ampoule consisted of a conical steel plug, a steel tube, and a steel capsule assembled coaxially. The conical steel plug and the capsule were made of low-carbon steel; the tube was made of 08Cr18Ni10Ti steel. A diagram of the ampoule is shown in Figure 2b. The outlet hole, which was intended for the exit of the reaction gases, was at the bottom of the capsule.

### 2.3. Explosive Parameters

Figure 3 shows a diagram and a photograph of the explosive setups. Ammonite and a mixture of microporous ammonium nitrate and diesel oil (ANFO) were used in the experiments. Table 1 lists explosive parameters.

### 2.4. Calculation of the Tube Collapse Parameters

The tube collapse velocity *υ* is determined from the equation below [35] (p. 225):(1)$$\upsilon =\frac{D}{2}\sqrt{\frac{r}{2\left(1+2r/\psi \right)}}$$
where *D* is the detonation velocity, m/s; *r* is the explosive ratio (explosive mass /tube mass ratio); and *ψ* equals 4 (for tube configuration).

The following equation was used to define explosive pressure (*P*) on the tube wall:(2)$$P=\frac{\rho \cdot D^{2}}{4}$$
where *ρ* is the explosive density, kg/m^3^.

The reduction ratio of the outer diameter of the tube was calculated using the following equation:(3)$$\eta =\frac{d_{0}-d_{1}}{d_{0}}\cdot 100\%$$
where *d_0_* is the initial outer diameter of the tube, and *d_1_* is the outer diameter of the tube after SWE.

### 2.5. Metallographic Analysis

Metallographic analysis of the longitudinal and transverse cross-sections of the metallographic specimens was carried out using a METAM LV-34 optical microscope and a TS-500 camera. The metallographic specimens were prepared using a metallographic grinding and polishing machine (ShLIF-1M/V) using diamond paste. SEM/EDS analysis was conducted with a Zeiss Ultraplus microscope equipped with an INCA 350 Oxford accessory. In this study, microhardness (HV) was measured using a PMT-3 Vickers hardness tester and MMS software. Loads of 100 g were applied for 15 s. X-Ray diffraction (XRD) analysis was carried out using DRON-3M diffractometer. The samples were scanned using Cu-Kα radiation, 20° to 80° (2Θ) with a scanning step of 0.02°, and exposure time of 1 s.

## 3. Results

### 3.1. SWE Experiments

Table 2 lists the different tube collapse parameters that were calculated in Equations (1)–(3).

Figure 4 shows ampoules No. 1 and No. 2 after the SWE experiments. The ampoules were studied in the following order: segment I, segment II, and segment III (see Figure 4). Arrow D shows the detonation direction from the detonator to the end of segment II.

### 3.2. Study of Segment I

Metallographic analysis of segments I of ampoules No. 1 and No. 2 revealed cracks formed as a result of the tube collapse (Figure 5a,b).

Figure 6 shows the bar charts of microhardness distribution in the transverse cross-sections. The microhardness measurement started at point 0 from the inner wall of the tube (point 0). The measurement results show that the average microhardness in both ampoules was the same, and it increased by 60–65% after SWE (Figure 6). The minimum microhardness in both ampoules was 325 HV, and the maximum was 550 HV.

### 3.3. Study of Segment II

After SWE, 53 vol.% of the initial powder sample was left in ampoule No. 1. The rest of the powder sample was driven into segment III. A chemical reaction was observed to have occurred only in the upper part of the powder sample (Figure 7a,b) and nowhere else. In addition, a new phase of the lower part was detected in the central region (Figure 7c,d).

In ampoule No. 2, 44 vol.% of the initial powder sample was left, and the synthesis took place in the entire volume of the powder sample (Figure 7e,f).

Figure 8a,b show the SEM image and the results of the EDS analysis of the upper part of segment II of ampoule No. 1. It was found that the upper part of the sample consisted of a NiAl phase containing up to 20 at.% Fe and 5 at.% Cr [36]. The lower part of the sample consisted of Ni–Al in the form of a compacted powder mixture. The synthesis products were detected only in the center of the sample in the region of 300 µm in diameter (Figure 8c). According to the EDS analysis results (Figure 8d), the new phase (in the central region) also consisted of NiAl and other elements (up to 10 at.% Cr and up to 40 at.% Fe).

Figure 8e,f show the SEM image of the lower part of ampoule No. 2 and the results of EDS analysis. It was found that the entire powder sample in segment II consisted of NiAl.

Studies of the interface between NiAl and the steel tube (ampoule No. 1) in the upper part revealed the presence of three structural regions (Figure 9 a). The first region is a transition region up to 5 µm wide, located between the tube wall and NiAl. This region consists of Ni_3_Al + 40 at.% Fe (Table 3, points 4 and 6). The second region is about 20 μm wide and consists of NiAl + 10–27 at.% Fe (Table 3, points 2, 3, 7, 8). The third region is about 70 μm wide and also consists of NiAl + 6 at.% Fe (Table 3, point 1). Thus, Fe content decreases from the interface deep into NiAl.

Table 4 shows the EDS analysis results of the interface between NiAl and the steel tube (ampoule No. 2). It shows that the region interface has a two-phase composition and consists of Ni_5_Al_3_ + Ni_3_Al.

Figure 10 shows a longitudinal section of segment II from ampoule No. 1. NiAl was synthesized to a depth of 5–7 mm. The transition region between the synthesized and the compacted part of NiAl is cone-shaped with an angle of α = 104° at the apex (Figure 10).

The EDS analysis of the phase composition of the transition region between the synthesized and compacted part in the sample of ampoule No. 1 showed that the transition region is divided into two parts (Figure 11a). The central part consists of NiAl with Fe content up to 20 at.% and Cr up to 7 at.% (Figure 11a,b, points 8 and 9) with width ranging from 0.2 to 6 mm. The other part of the powder sample consists of NiAl_3_ with a maximum width of 100 µm (Figure 11a,b, points 7, 10).

### 3.4. Study of Segment III

According to the explosive setups, the capsules in both ampoules were not subjected to SWL; they were necessary only for NiAl accumulation. Both capsules essentially yielded the same results (Figure 12a). Namely, some of the pressed powder flew out through the outlet hole (some traces of NiAl were found on the walls of the hole); some of the NiAl was also found on the inner walls of the capsule. Only a small portion of NiAl was in the lower part of the capsule. The XRD results show that this NiAl has a single-phase NiAl composition (Figure 12b).

## 4. Discussion

The results of this study demonstrate that the new SWE technology makes it possible to produce a composite material with a NiAl layer. The general principle underlying this technology consists of carrying out intensive multi-level deformation of a powder sample using SWL and its pressing through a hole with a smaller diameter.

Two SWE modes, which differed in detonation velocity, tube collapse velocity, the r parameter, and the pressure behind the shock wave front, were used. It was found that the higher the SWE parameters, the less of the final material was driven into the capsule (47 vol.%). Consequently, the lower parameters produced more final material in the capsule (56 vol.%). In addition, a higher velocity of collapse and detonation caused the early compaction of the powder sample in its lower part, which led to the collapse of the void space and internal pores and moved the air up along the axis of the powder sample. As a result, a large central pore formed, which took up about 25% of the total area (Figure 10). The increase in size of the central pore stopped the reaction of exothermic synthesis. This halt resulted in the quenching of the reaction products [37,38,39].

The results of the experiments can be explained by several circumstances: lower velocity and pressure created conditions for a longer exposure of the explosive on the tube walls, which ensured the pushing of a larger mass of powder sample into the capsule. The longer exposure also led to a greater reduction ratio of the outer diameter of the tube (32%) than in the case of a faster exposure (27%). In addition, a larger volume of the final product was achieved by increasing the diameter of the central hole in the initial powder sample from 2.5 to 5 mm. Based on this, we can conclude that to increase the volume of the final product in the capsule, it is necessary to reduce the tube collapse velocity and the detonation velocity and also to change the configuration of the powder sample to ensure a quicker collapse of the sample wall during SWE. Changing the configuration of the powder sample may involve making a cone-shaped central hole in the lower part of the powder sample. The findings of the study should be considered preliminary and require further detailed theoretical and experimental research.

It is well-known that in the case of using a cylindrical ampoule for the SWL of a powder sample, the Mach stem is formed in the center of the sample due to the accumulation of shock-wave energy [23]. As a result, extremely high pressure and temperature are exerted, which leads to inhomogeneity of the final product structure. However, the present study shows that the central through hole (2.5 mm in diameter) in the powder sample partially or completely negates the effect of the Mach stem on the homogeneity of the NiAl structure. Moreover, in the powder sample with a central through hole of 5 mm in diameter, the effect of the Mach stem was not observed at all.

The results of the metallographic analysis indicate that the process of high-velocity collapse of the tube in ampoule No. 1 created conditions for the formation of cumulative flow, which directly affected the phase formation in NiAl (Figure 7c, Figure 8c and Figure 11a,b). Presumably, this cumulative flow formed during the tube collapse when the deformation of the inner wall and its surface activation occurred, contributing to the capture and removal of metal particles from the surfaces by the cumulative flow.

Figure 10 shows a cone-shaped transition region that formed between the synthesized and compacted parts of the powder sample in ampoule No.1. The EDS analysis showed that this region consists of transition intermetallic compounds, namely, NiAl_3_, Ni_2_Al_3_ and Ni_3_Al (Figure 11). The synthesized part has a complex elemental composition consisting of NiAl, Fe and Cr particles introduced in NiAl by the cumulative flow. Meanwhile, no elements of the cumulative flow (Fe or Cr) were found in ampoule No. 2. The analysis of the capsule inner surface (section III) showed the presence of a thin layer of NiAl on the capsule inner walls, which indicates that intense dispersion of liquid NiAl took place in SWE.

The findings of the study should be considered preliminary and require further detailed theoretical and experimental research.

## 5. Conclusions

The present study involved two experiments for producing a composite metal–intermetallic material by using a novel complex method consisting of SWL, SHS and extrusion. The proposed name for this novel method is shock-wave extrusion (SWE). The main advantage of this method is the use of a one-stage operation without mechanical activation and heating of the powder mixture. As a result, it is possible to produce a composite material with an outer metal layer (steel, titanium, aluminum, etc.) and an inner layer of material with high hardness and heat resistance (cermets, intermetallic compounds, etc.). In addition, the intensive multi-level deformation of the powder sample enables modifications of the structure of the inner layer. It is worth noting that SWE is complex in nature and is not yet fully understood. This caused some difficulties in the experimental setup’s design. The goals of further studies are achieving 90–100 vol.% of the final product and reducing the porosity in the final product.

## Figures and Tables

**Figure 1 materials-15-06062-f001:**
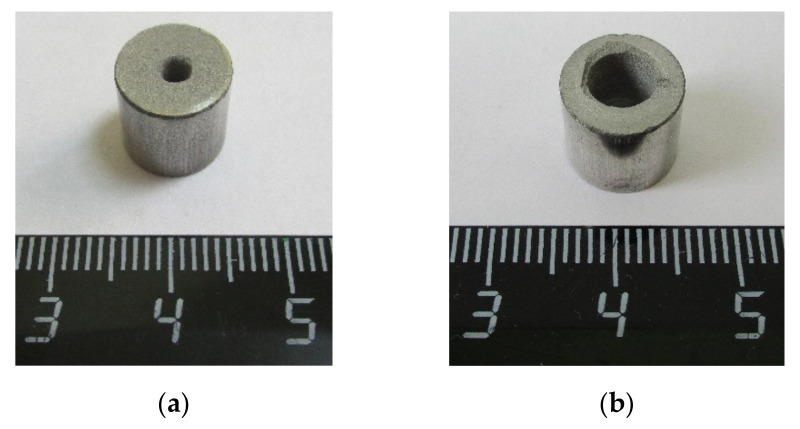
Ni-Al powder cylindrical samples: (**a**) for ampoule No. 1; (**b**) for ampoule No. 2.

**Figure 2 materials-15-06062-f002:**
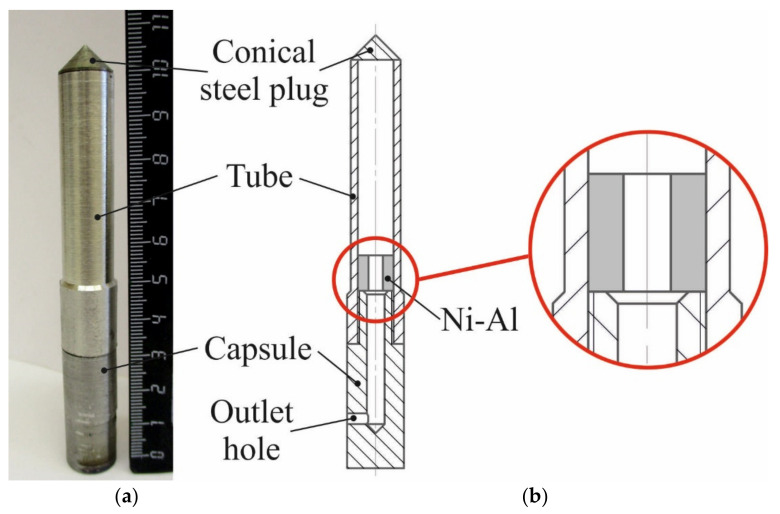
(**a**) Photograph of the cylindrical ampoule; (**b**) diagram of the cylindrical ampoule.

**Figure 3 materials-15-06062-f003:**
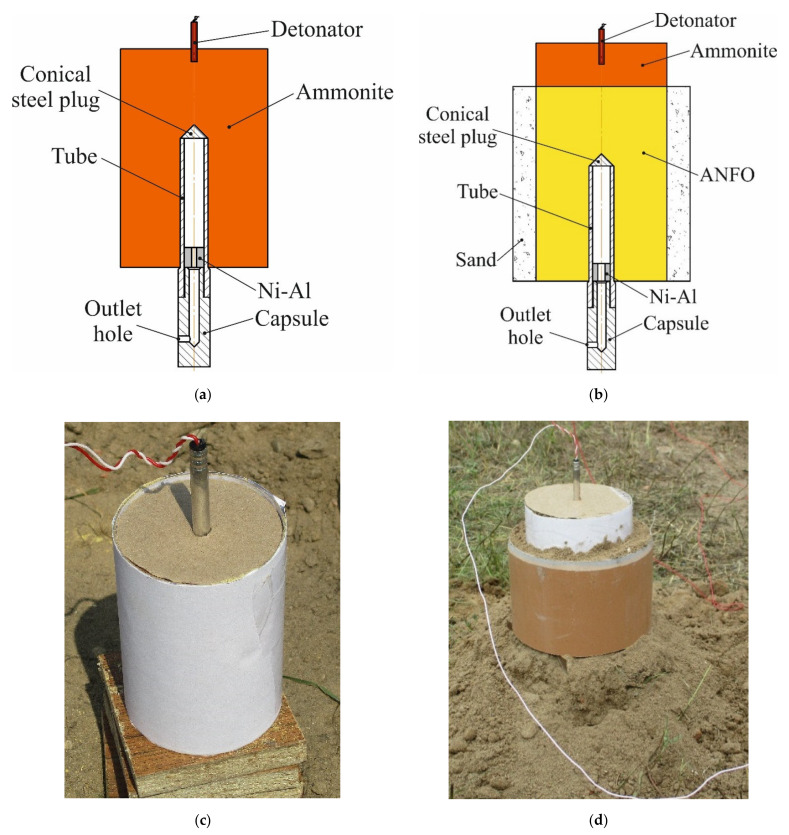
(**a**,**b**) Diagram of explosive setups; (**c**,**d**) photograph of explosive setups.

**Figure 4 materials-15-06062-f004:**
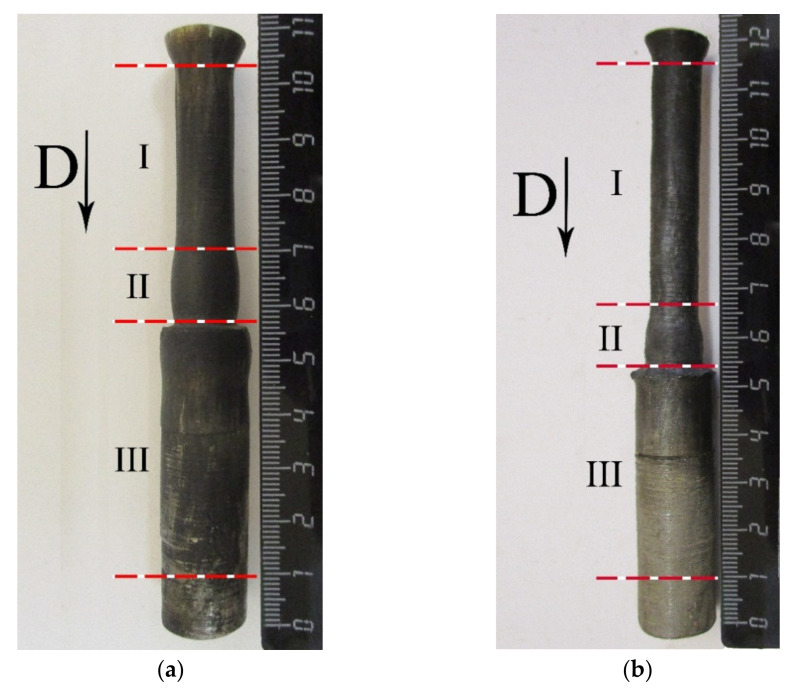
Photograph of ampoules after SWE and segments: (**a**) ampoule No. 1; (**b**) ampoule No. 2.

**Figure 5 materials-15-06062-f005:**
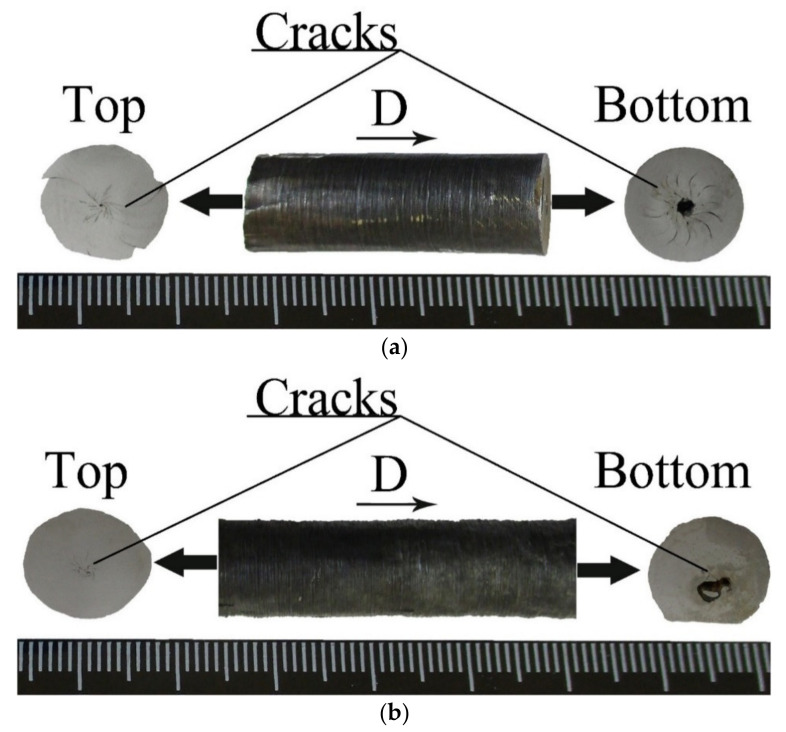
Photograph of segment I: (**a**) ampoule No. 1; (**b**) ampoule No. 2.

**Figure 6 materials-15-06062-f006:**
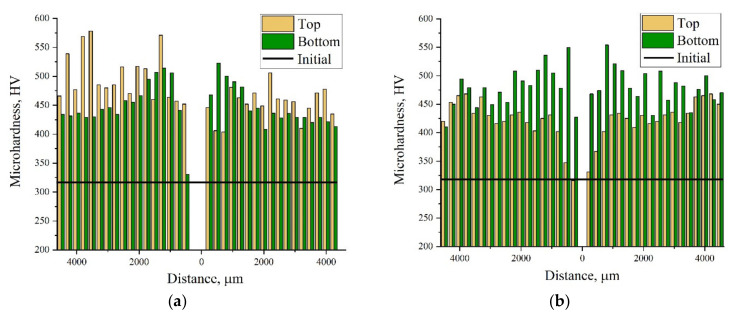
Bar charts of microhardness distribution in segment I: (**a**) ampoule No.1; (**b**) ampoule No. 2.

**Figure 7 materials-15-06062-f007:**
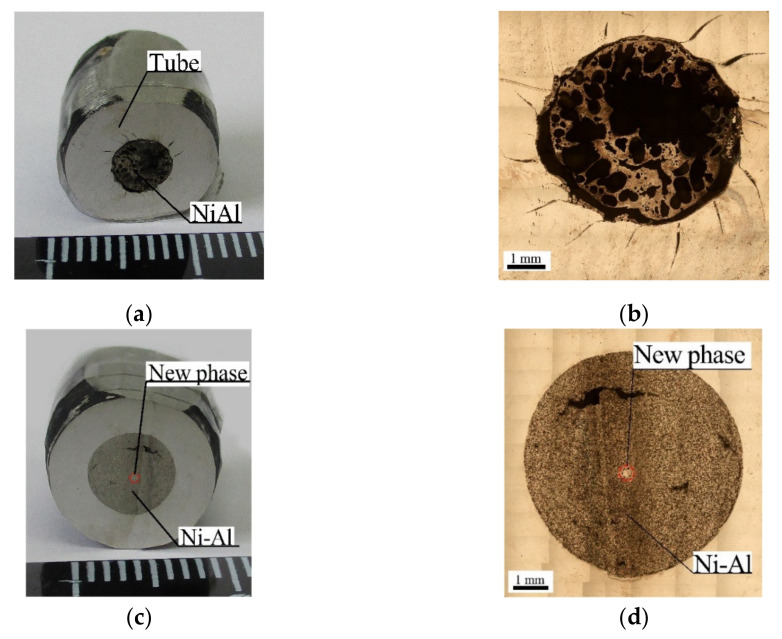
Photographs of structures of segment II: (**a**,**b**) upper part (ampoule No. 1); (**c**,**d**) lower part (ampoule No. 1); (**e**,**f**) lower part (ampoule No. 2).

**Figure 8 materials-15-06062-f008:**
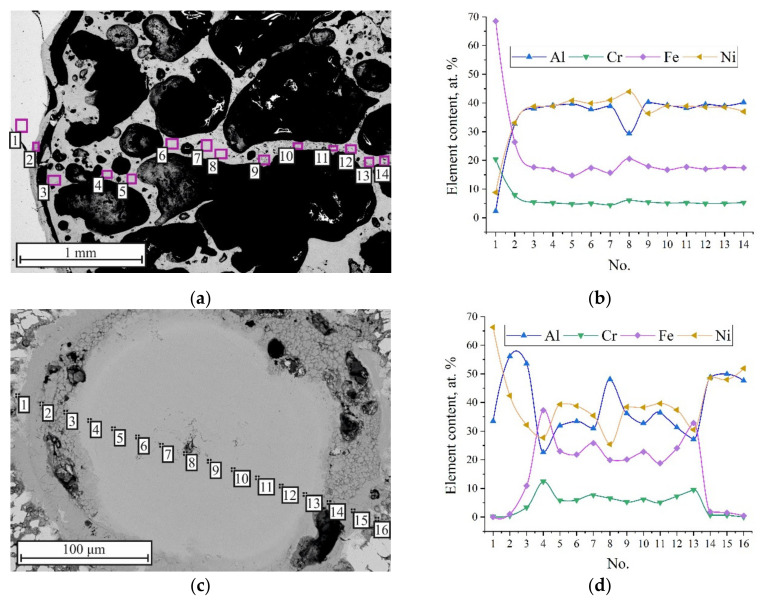
SEM images of segment II microstructure and EDS analysis results: (**a**) upper part in ampoule No. 1; (**c**) lower part in ampoule No. 1; (**e**) lower part in ampoule No. 2; (**b**,**d**,**f**) element distribution after EDS analysis.

**Figure 9 materials-15-06062-f009:**
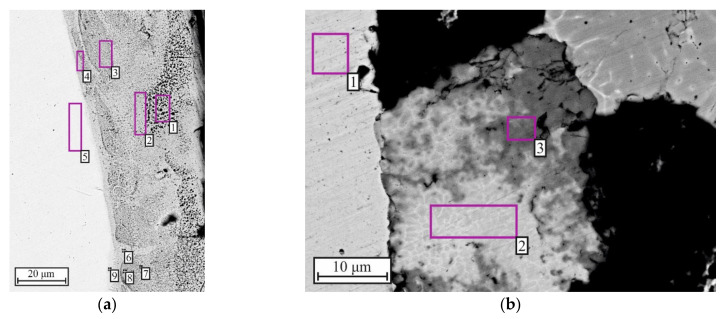
SEM image of the microstructure of the tube and sample interface in segment II: (**a**) ampoule No. 1; (**b**) ampoule No. 2.

**Figure 10 materials-15-06062-f010:**
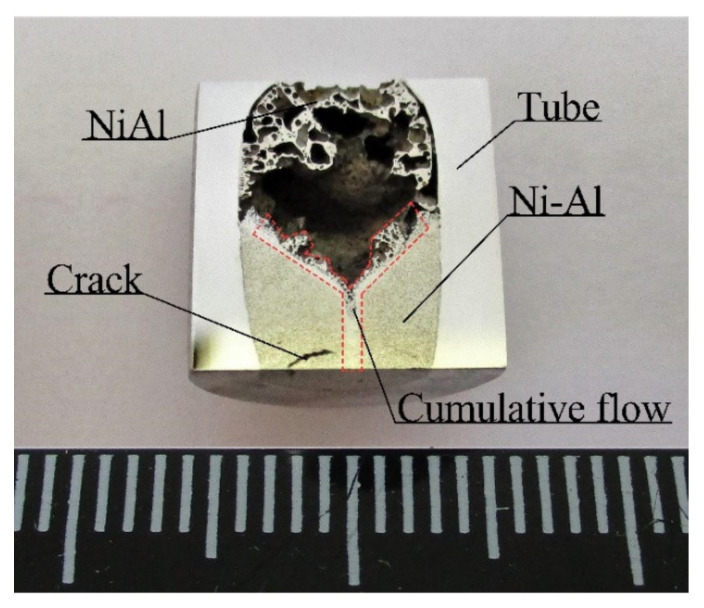
Photograph of longitudinal section from segment II of ampoule No. 1.

**Figure 11 materials-15-06062-f011:**
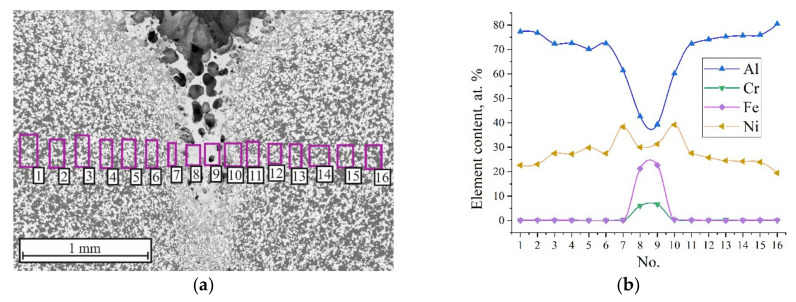
SEM image of transition region in the powder sample (segment II): (**a**) SEM image of central part; (**b**) element distribution after EDS analysis (from point 1 to point 16).

**Figure 12 materials-15-06062-f012:**
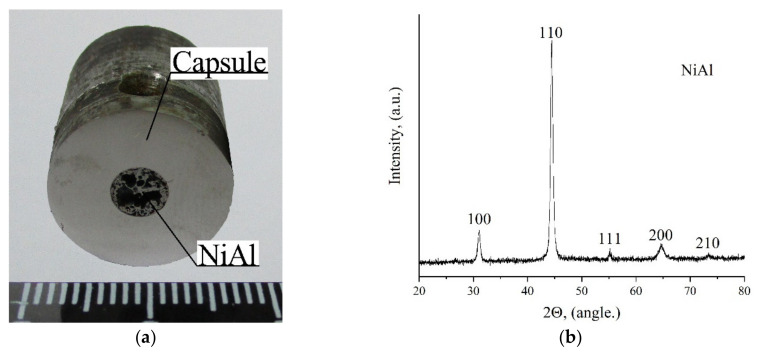
(**a**) Photograph of segment III (ampoule No.1); (**b**) XRD pattern of NiAl powder after SWE.

**Table 1 materials-15-06062-t001:** Explosive parameters.

Ampoule Number	Explosive	Detonation Velocity D, m/s	Explosive Ratio, r *
1	Ammonite	3750	1.77
2	ANFO	3150	2.45

* Explosive mass/tube mass.

**Table 2 materials-15-06062-t002:** The tube collapse parameters.

Ampoule Number	The Tube Collapse Velocity υ, m/s	The Explosive Pressure on the Tube Wall P, GPa	Reduction Ratio of Outer Diameter of the Tube, %
1	1290	3.5	27
2	1170	1.9	32

**Table 3 materials-15-06062-t003:** EDS analysis results for ampoule No. 1 in according to each point marked in Figure 9a.

Element Content, at.%				
No.	Cr	Fe	Ni	Al
1	2.0	6.0	46.9	45.1
2	7.0	23.3	37.6	32.1
3	5.6	18.1	38.5	37.8
4	14.1	43.0	24.5	18.4
5	19.7	68.4	9.0	2.9
6	13.3	43.8	25.6	17.3
7	9.5	27.3	31.3	31.9
8	3.3	10.8	43.3	42.6
9	16.6	54.5	15.5	13.4

**Table 4 materials-15-06062-t004:** EDS analysis results for ampoule No. 2 in according to each point marked in Figure 9a.

Element Content, at.%				
No.	Cr	Fe	Ni	Al
1	19.9	70.3	9.8	-
2	-	-	67.6	32.4
3	-	-	43.7	56.3

## Data Availability

Not applicable.

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
