# Peer review of "Synthesis of NiAl Intermetallic Compound under Shock-Wave Extrusion"

_materials, 2022, doi:10.3390/ma15176062_

Round 1

Reviewer 1 Report

This work is focused on the synthesis of NiAl intermetallic compound under shock-wave extrusion and it is an interesting work; however; there are several points that need to be clarified:

1) The figure captions must be improved because they are not understable.

2) Some experimental details  are missing for example,  XRD radiation, etchant.

3) Since the NiAl intermetallic compound is an ordered phase,  the XRD pattern must show the Miller index of each plane.

4) Is the average hardness the same for the two specimens? If not, expain the differences.

5) The micrograph quality is very poor, higher magnification micrographs  are needed to show the microconstituents in each region. For instance, the NiAl is alone or embeden in a Ni-rich matrix. What is the new phase mentioned in the results?

6) The combination of figures and tables is confused. Please, separate them. The same designation is used 1,2, 3...

7) It is important to relate the results of harndness to microstructure.

8) Can the Fe and Cr contamination be avoided?

9) The differences between specimens 1 and 2 can be summarized in a Table.

Reviewer 2 Report

In this manuscript, the author reported the synthesis of NiAl intermetallic compound under shock-wave extrusion. The proposed method is promising and the authors' presented it very well. The explanation is good. Hence, I am accepting the manuscript as it is.

Few point to authors

It would be better to talk about the status of using this method to make other intermetallic compounds.

The authors only reported on the NiAl intermetallic compound. What about the Ni3Al and NiAl3 intermetallics? Is this the method applicable for these intermetallic compounds?

Reviewer 3 Report

The manuscript "Synthesis of NiAl intermetallic compound under shock-wave extrusion" has been reviewed.

It deals with the production of composite metal-intermetallic materials by shock-wave extrusion.

Preliminary results deriving from n. 2 experiments are presented. However this aspect should be better evidenced in the text, in particular that these results are preliminary and need deeper investigations in different experimental conditions.

By the way, the manuscript is interesting, almost well organized. However english should be revised by a native speaker.

I don't have any particular concerns, in my opinion it can be considered for publication after the following minor revisions.

Line 20: please check "90100%".

Line 30: please check [36], [3-6]?

Line 85: и translate.

Line 90: please summarize the results of [34].

Line 112: please check.

Line 134: process.3.2. Figures, Tables and Schemes. Please check.

Line 198: According to the of explosive setups. Please check.

Line 223: please check [3739], [37-39]?

Ref. 35 is not called in the main text.

Round 2

Reviewer 1 Report

None